There are amendments to this paper

# Identification of recurrent *USP48* and *BRAF* mutations in Cushing's disease

Jianhua Chen et al.[#]

Cushing's disease results from corticotroph adenomas of the pituitary that hypersecrete adrenocorticotropin (ACTH), leading to excess glucocorticoid and hypercortisolism. Mutations of the deubiquitinase gene *USP8* occur in 35–62% of corticotroph adenomas. However, the major driver mutations in *USP8* wild-type tumors remain elusive. Here, we report recurrent mutations in the deubiquitinase gene *USP48* (predominantly encoding p.M415I or p.M415V; 21/91 subjects) and *BRAF* (encoding p.V600E; 15/91 subjects) in corticotroph adenomas with wild-type *USP8*. Similar to *USP8* mutants, both *USP48* and *BRAF* mutants enhance the promoter activity and transcription of the gene encoding proopiomelanocortin (POMC), which is the precursor of ACTH, providing a potential mechanism for ACTH overproduction in corticotroph adenomas. Moreover, primary corticotroph tumor cells harboring *BRAF* V600E are sensitive to the BRAF inhibitor vemurafenib. Our study thus contributes to the understanding of the molecular mechanism of the pathogenesis of corticotroph adenoma and informs therapeutic targets for this disease.

Correspondence and requests for materials should be addressed to Y.Z. (email: zhaoyaohs@vip.sina.com) or to C.H. (email: huangcx@shsmu.edu.cn) or to Y.S. (email: shiyongyong@gmail.com). [#]A full list of authors and their affliations appears at the end of the paper.

Cushing's disease is caused by the hypersecretion of adrenocorticotropin (ACTH) from pituitary corticotroph adenomas[1]. Chronic elevation of ACTH stimulates the adrenal glands to secrete excessive glucocorticoids, which subsequently induces hypercortisolism[2,3]. Cushing's disease is a severe disease, and when left untreated, patients may develop cardiovascular disease, catabolic symptoms, hypersensitivity to infections, and mood disorders. Cushing's disease has a prevalence of 39 cases per million of population[4,5]. The diagnosis of hypercortisolism and preoperative localization of the adenoma in this disease are complicated and sometimes difficult. Currently, the first-line treatment for this disease is pituitary adenomectomy. Only 65–90% of patients achieve complete or partial remission after initial transsphenoidal surgery[6]. A substantial proportion of patients tend to recur after a period of remission. It is ineffective in the treatment of patients with residual excess cortisol due to recurrent tumors or incomplete removal of tumor tissues, and no targeted therapy is currently available. So far, there has been limited genome-level study of mutations in corticotroph adenomas, hampering the development of diagnostic and therapeutic approaches for Cushing's disease.

Recently, the deubiquitinase gene USP8 was found to be mutated in 35–62% of corticotroph adenomas, resulting in sustained activation of the epidermal growth receptor (EGFR), the mitogen-activated protein kinase (MAPK) pathway, and subsequent overproduction of ACTH[7,8]. Inhibition of the USP8 pathway has been suggested as a targeted therapeutic strategy for the treatment of patients with USP8-mutated adenomas[9,10]. Exome-sequencing studies also revealed some potential genetic lesions closely related to the pathogenic mechanism in USP8 wild-type corticotroph adenomas, including loss-of-function mutations of the glucocorticoid receptor gene NR3C1[7,8,11], a critical negative regulator of ACTH production. However, these studies included only a small number of corticotroph adenomas with wild-type USP8. The genetic basis for the development of USP8 wild-type corticotroph adenomas has not yet been fully identified.

In this report, we reveal recurrent mutations in the deubiquitinase gene USP48 and BRAF in corticotroph adenomas with wild-type USP8. Similar to USP8 mutants, both USP48 and BRAF mutants enhance the promoter activity and transcription of the gene encoding proopiomelanocortin (POMC), which is the precursor of ACTH. Our study thus contributes to the understanding of the molecular mechanism of the pathogenesis of corticotroph adenoma and informs therapeutic targets for this disease.

## Results

**Somatic USP48 and BRAF mutations in Cushing's disease**. To gain further insight into the molecular mechanisms of the pathogenesis of USP8 wild-type corticotroph adenomas, we conducted exome-sequencing of 22 paired tumor tissues and peripheral blood samples. The mean sequencing depth was 131× for the tumor samples and 95× for matched blood samples, with >10× coverage for 93.9% of tumor samples and 94.4% of blood samples (Supplementary Figure 1). We observed a preference for C > T/G > A alterations analogous to the somatic single-nucleotide variation (SNV) spectrum (Fig. 1a). Next, we used deconstructSigs[12] to identify the mutational signatures of our samples. Mutation Signature 1A is associated with the SNV spectrum found in these tumors, characterized by the prominence of C > T substitutions at the NpCpG trinucleotides[13]. CpG dinucleotide is the main mutational hot-spot in most human cancers. Some C > T mutations occurred at the CpG sites. We identified a total of 144 candidate non-silent somatic mutations among the 22 cases, including 115 missense, seven nonsense, five splice-site, and 17 insertion or deletion (indel) mutations, resulting in an average of 7

(ranging from 2 to 34) somatic mutations in the exon regions (Fig. 1b and Supplementary Data 1).

From this analysis, we identified four recurrently-mutated genes, including BRAF, the deubiquitinase gene USP48, NR3C1, and host cell factor C1 (HCFC1) (Fig. 1b and Supplementary Table 1). NR3C1 mutation has previously been discovered in corticotroph adenomas[7,14]. BRAF and USP48 mutations have never been reported to be related to this disease yet. We validated these recurrent mutations by Sanger sequencing (Supplementary Figure 2). Six of the 22 tumors had mutations in USP48 (four encoding p.M415I and two encoding p.M415V), and two of the 22 cases had BRAF mutations encoding p.V600E (Supplementary Figure 3). We next examined the prevalence of USP48 and BRAF mutations in additional corticotroph adenomas with wild-type USP8 by targeted sequencing. Combining the data from whole-exome and targeted sequencing, which included a total of 91 samples, we found that 16.5% of the cases (15/91) had BRAF V600E mutations and 23.1% of the cases (21/91) harbored USP48 mutations. Among the 21 USP48-mutated cases, 11 cases had the M415I mutation and four cases harbored the M415V mutation (Supplementary Table 2). In addition, we sequenced USP48 and BRAF in 78 corticotroph adenomas with mutated USP8. In contrast, only four cases had BRAF mutations (5.1%) and one case had a USP48 mutation (1.2%) (Fig. 1c). Within a total of 169 cases, USP8 mutations are exclusive from mutations in either USP48 or BRAF in all the 169 cases ($P = 1.208 \times 10^{-5}$ and $P = 0.027$ receptively, two-tailed Fisher's exact test). However, USP48 and BRAF mutations are not mutually exclusive (Fig. 1c) ($P > 0.05$, two-tailed Fisher's exact test). We next sought to investigate whether similar mutation patterns were also present in other types of pituitary adenomas. Strikingly, none of the samples, including 20 growth hormone-secreting, 20 prolactin-secreting, and 20 non-functioning adenomas, carried either USP48 or BRAF mutations (Supplementary Figure 3). Therefore, similar to USP8, BRAF and USP48 mutations appear to be unique genetic signatures of corticotroph adenomas.

**Functional characterization of BRAF V600E mutants**. It has been well established that BRAF V600E results in constitutive activation of the BRAF kinase activity and its downstream MAPK pathway in many types of human cancers[15,16]. BRAF is indispensible for c-AMP dependent MAPK activation. However, it has not been detected in AtT-20 cells. When we ectopically expressed BRAF V600E in a murine cell line of corticotroph adenoma (AtT-20), we observed elevated phosphorylation of Erk1/2, an indicator of MAPK activation, compared to cells with wild-type BRAF expression (Fig. 2a). In addition, immunohistochemical analysis demonstrated that corticotroph adenomas carrying the BRAF V600E mutation had higher MAPK activity, as indicated by increased phosphorylation of Erk1/2 in these tumors compared to wild-type cases (Fig. 2b).

We next explored critical Erk1/2 targets that could mediate the effect of BRAF V600E. Nur77 is a key transcription activator of POMC through direct binding to the Nur77-binding response element (NBRE) and Nur77 response element (NurRE) in the POMC promoter region[17–19] (Supplementary Figure 4). In addition, POMC transcription can also be induced by the c-Jun/c-Fos heterodimer, which acts by binding to the canonical AP1 binding site in the first exon of POMC[20,21] (Supplementary Figure 4). Activation of the MAPK pathway could stimulate the phosphorylation and transcriptional activation of the transcriptional regulators Nur77[22], c-Jun, and c-Fos[23]. As expected, we revealed that BRAF V600E expression induced phosphorylation of Nur77, c-jun, and c-fos in the AtT-20 cells (Fig. 2c). Immunohistochemical analysis further proved that

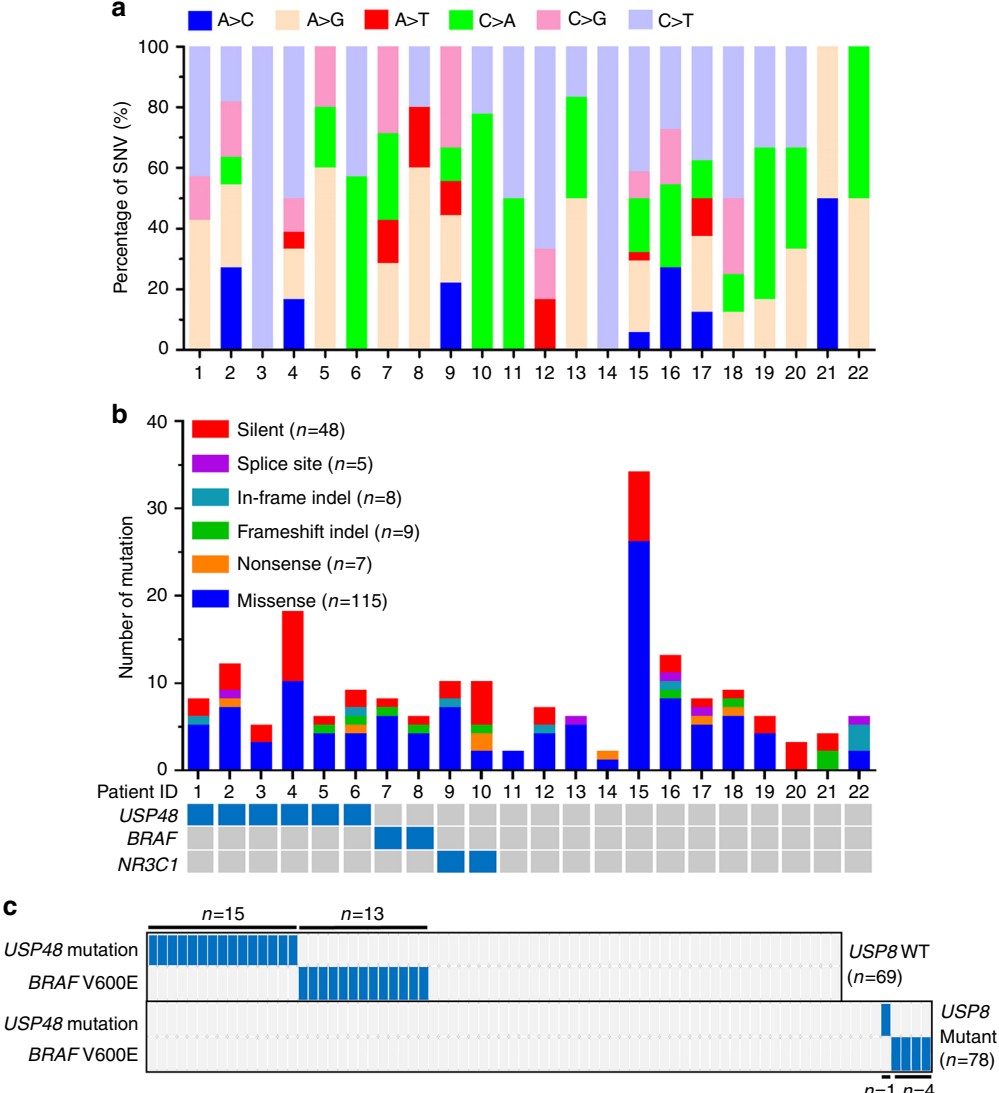

**Fig. 1** Whole-exome sequencing and targeted sequencing in corticotroph adenomas. **a** Percentage of non-silent somatic SNVs identified by whole-exome sequencing in 22 subjects with corticotroph adenomas carrying wild-type *USP8*. **b** The number and type of somatic mutations identified by whole-exome sequencing (top). The mutational status of the indicated genes in each patient is shown at the bottom. **c** Mutation frequency of the indicated genes in corticotroph adenomas with wild-type or mutated *USP8* by targeted sequencing. Each column represents a single case. *n*: number of subjects. Patients with the indicated mutations are marked in blue (**b, c**)

*BRAF*-mutated, but not wild-type corticotroph adenomas were positive for phosphorylation of Nur77, c-jun, and c-fos (Supplementary Figure 5).

Next, using luciferase reporter assays, we showed that the introduction of wild-type *BRAF*, as compared to the vector control, resulted in stimulatory activity of the *POMC* promoter, and *BRAF* V600E obviously potentiated this effect (Fig. 2d). Furthermore, the stimulatory effect of *BRAF* V600E on activities of the NurRE-, NBRE-, or AP1-mutated *POMC* promoters was considerably impaired (Fig. 2d). Finally, ectopic expression of *BRAF* V600E in AtT-20 cells resulted in a substantially increased level of *POMC* mRNA compared to cells with wild-type *BRAF* (Fig. 2e). Compared to tumor cells with wild-type *BRAF*, tumor cells with *BRAF* V600E displayed a greater reduction of ACTH secretion in response to vemurafenib, with an approximate half-maximal inhibitory concentration (IC50) of 0.3 μM, within the range of IC50 values reported for vemurafenib-sensitive tumor cells (Fig. 2f). Moreover, vemurafenib treatment did not cause more cell death in tumors with *BRAF* V600E than wild-type

(Supplementary Figure 6). These results supported the potential efficacy of vemurafenib in the treatment of corticotroph adenomas with the *BRAF* V600E mutation. Taken together, these results indicate that *BRAF* V600E contribute to the pathogenesis of ACTH-secreting pituitary adenomas through increased phosphorylation of *POMC* transcription regulators including Nur77, c-jun, and c-fos.

**Functional characterization of *USP48* mutants.** Missense mutations of *USP48*, including M415I/V substitutions, were also identified as most frequent mutations in *USP8* wild-type corticotroph adenomas. Immunohistochemical analysis of USP48 protein in a series of corticotroph adenomas revealed that there was no degradation or enhanced expression of *USP48* in *USP48*-mutated tumors compared to tumors with wild-type *USP48* (Supplementary Figure 7). The M415 residue was mapped to the peptidase domain of USP48 protein (Fig. 3a) and at well-conserved amino acid positions across distinct species (Supplementary Figure 8).

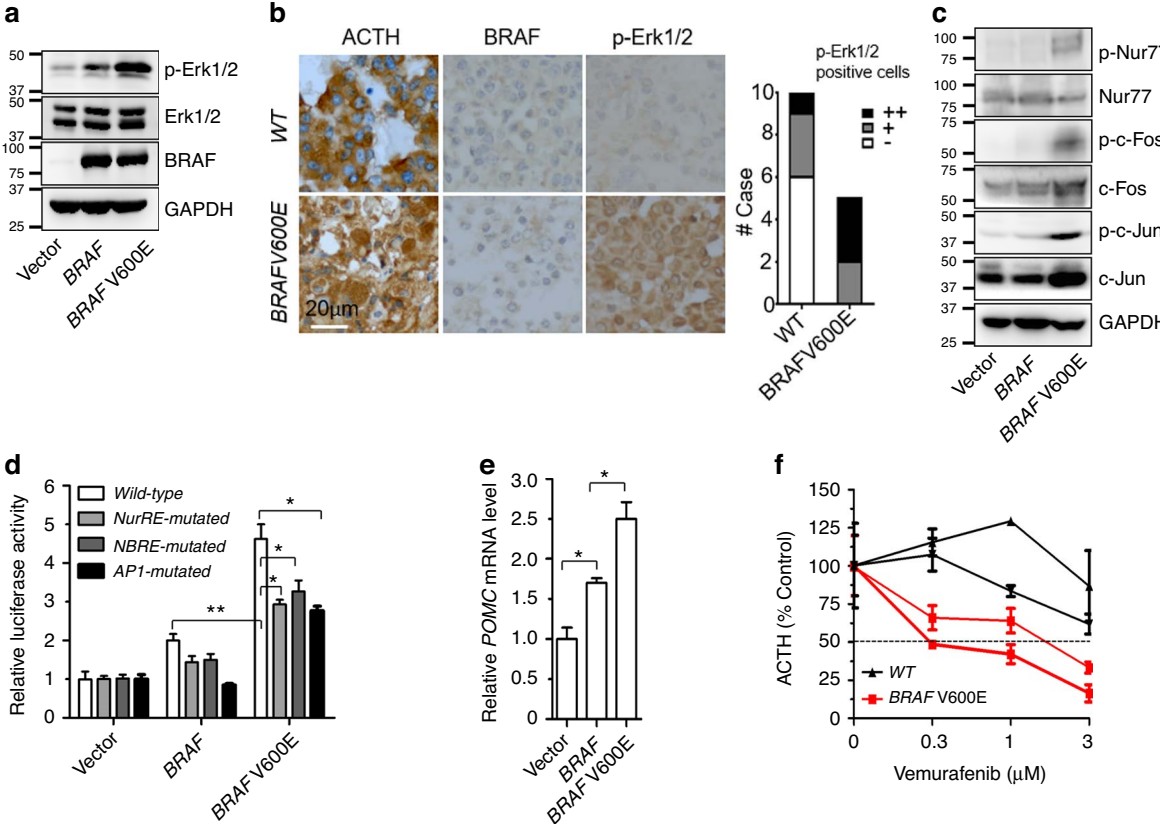

**Fig. 2** *BRAF* V600E promotes *POMC* promoter activity and ACTH production. **a** Immunoblotting analysis of p-Erk1/2 and total Erk1/2 in AtT-20 stably infected with wild-type *BRAF* or *BRAF* V600E. **b** Immunohistochemical study of p-Erk1/2 in tumor samples according to the mutational status of *BRAF*. Scale bars, 20 μm. *P*-value was calculated using two-tailed Fisher's exact test (*P* = 0.036). **c** Immunoblotting analysis of indicated proteins in AtT-20 cells as described in **a**. **d** Activities of wild-type or various mutated *POMC* promoters in cells transfected with wild-type *BRAF* or *BRAF* V600E together with a control vector expressing renilla luciferase. *Y*-axis: ratio of luciferase activity/renilla activity, presented as the means of three experiments and expressed relative to empty control vector. Error bars represent SEM of three measurements. **e** Relative mRNA levels of *POMC* in AtT-20 cells stably infected with indicated constructs. Data are expressed relative to the empty control vector after normalization to Hprt. Error bars represent standard SEM of three measurements. *$*P < 0.05$, $**P < 0.01$, two-tailed Student's *t*-test (**d, e**). **f** The inhibitory effect of vemurafenib on ACTH production in primary cultured human corticotroph adenomas (two wild-type and two *BRAF* V600E). ACTH levels in culture media were measured by radioimmunoassay and presented as % of the control media. Error bars indicate SEM of three replicates. The dashed line represents $IC_{50}$

Molecular modeling further revealed that the M415 residue was spatially located in the catalytic palm of USP8 protein and inferred to be involved in regulating its catalytic activity (Fig. 3b). We next searched for the molecular mechanisms underlying *USP48* mutation-mediated *POMC* transcription. We first examined whether *USP48* mutants activate similar signaling pathways as *BRAF* V600E. Western blot analysis showed that the M415I/V mutants were not able to promote phosphorylation of Erk1/2, Nur77, c-jun, and c-fos (Supplementary Figure 9). Immunohistochemistry also proved that *USP48*-mutated corticotroph adenomas were negative or weakly positive for the phosphorylation of these molecules (Supplementary Figure 5). Collectively, these results demonstrate that the molecular mechanisms underlying *USP48* mutation-mediated *POMC* transcription do not involve Erk1/2 or its downstream signaling cascade.

*USP48* has been implicated in the activation of NF-κB[24,25], which directly binds to the promoter of *POMC* and participates in regulating its transcription in AtT-20 cells[26]. We examined whether *USP48* mutation affects its interaction with RelA, a subunit of NF-κB. As shown in Supplementary Figure 10, the M415I/V mutants retained the ability to interact with RelA. Then we conducted luciferase reporter assays to evaluate the effect of M415I/V alteration on the activity of USP48 protein. *USP48* M415I/V mutants had a

greater inhibitory effect on the activity of NF-κB reporter than wild-type *USP48*, indicating gain-of-function of the M415I/V mutation (Fig. 3c). We also demonstrated that overexpression of wild-type *USP48* had a minor stimulatory effect on the *POMC* promoter activity compared to the control vector; however, this effect was markedly potentiated by either M415I or M415V mutants (Fig. 3d). Finally, the mRNA abundance of *POMC* was higher in AtT-20 cells stably transfected with *USP48* M415I/V than in cells with wild-type *USP48* (Fig. 3e).

**Clinical phenotypes and mutational status**. We previously reported that *USP8*-mutated corticotroph adenomas are small in size and have higher ACTH production. Thus, we explored whether patients with *BRAF* V600E or *USP48*-mutated tumor display distinct clinical features. Patients with *BRAF* V600E had significantly higher levels of midnight plasma ACTH (*P* = 0.023, Mann–Whitney *U*-test) and midnight serum cortisol (*P* = 0.007, Mann–Whitney *U*-test) (Table 1), but similar tumor sizes, when compared with patients carrying wild-type *BRAF/USP48*. Therefore, *BRAF* V600E appears to promote ACTH production. Other clinical features examined are undistinguished between patients with wild-type *BRAF* and *BRAF* V600E. Patients with *USP48*-mutated tumors have quite similar clinical features as those with wild-type *BRAF/USP48*.

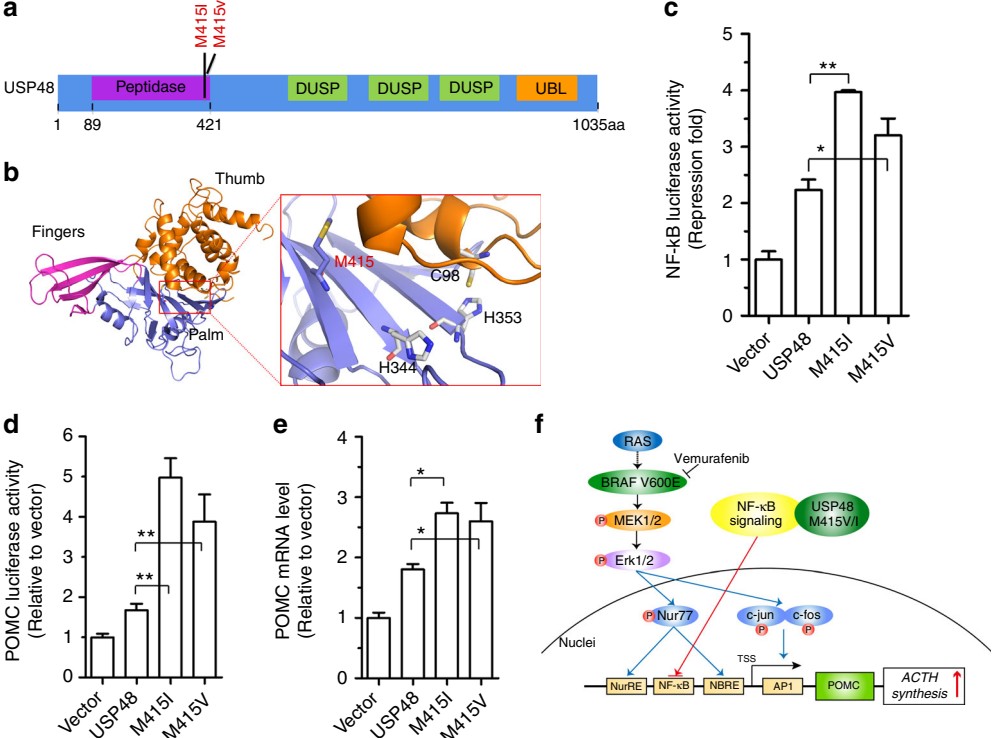

**Fig. 3** *USP48* mutants potentiate *POMC* promoter activity and ACTH production. **a** Schematic diagram of *USP48* domains and location of the M415 residue. DUSP, domain presents in ubiquitin-specific proteases; UBL, ubiquitin-like domain. **b** The modeling structure of peptidase domain of USP48. The crystal structure of USP7 (PDB ID: 1NBF) ubiquitin aldehyde (ubal)-binding form was used as the template to build the 3D structure of peptidase of USP48. The M415 residue is located in the catalytic palm, and three catalytic triads (C98, H344, and H353) are shown. **c, d** The NF-κB reporter (**c**) or *POMC* promoter (**d**) activities in cells transfected with indicated expression vectors together with control vectors expressing renilla and luciferase. Data are ratios of luciferase activity/renilla activity and presented as the mean of three experiments. **e** Relative *POMC* mRNA levels in AtT-20 cells stably infected with indicated expression constructs. Data are expressed relative to empty control vectors after normalization to Hprt. Error bars represent SEM of three measurements. *$P < 0.05$, **$P < 0.01$, two-tailed Student's *t*-test (**c–e**). **f** Schematic representation showing the mechanisms by which *BRAF* V600E and *USP48* mutants promote *POMC* transcription in corticotroph adenomas

## Discussion

Recently, two exome sequencing studies suggested the presence of mutations in the *USP8* genes of 35–62% of people with corticotroph adenomas[7,8]. However, the potential genetic risk of *USP8* wild-type corticotroph adenomas remains unknown. Here, we performed further exome sequencing and identified recurrently-mutated genes including *BRAF* and *USP48* in *USP8* wild-type corticotroph adenomas. Corticotroph adenomas showed a preference for C > T mutation, and some of these alterations occurred at the CpG sites. The C of CpG is a preferential site of methylation, and the methyl-C is prone to spontaneous deamination to T[27]. This is the most likely reason for CpG dinucleotide to be the main mutation hot-spot in most human cancers. Therefore, it is not surprising that corticotroph adenomas have this mutation spectrum. These results provide a comprehensive genetic landscape for this tumor. We found that mutations in *USP48* and *BRAF* were also key genetic lesions of Cushing's disease. Six of the 22 tumors had mutations in *USP48* (p.M415I or p.M415V) and two of the 22 cases had *BRAF* mutations (p.V600E) by exome sequencing. We screened the prevalence of *USP48* and *BRAF* mutations in additional 69 corticotroph adenomas with wild-type *USP8*. We found that 18.8% of these cases (13/69) had *BRAF* V600E mutations and 21.7% of these cases (15/69) harbored *USP48* mutations. However, neither *BRAF* V600E nor *USP48* M415I/M415V was detected in other types of pituitary adenomas. Therefore, similar to *USP8*, *BRAF* and *USP48* mutations appear to be unique genetic signatures of corticotroph adenomas. We

assumed that mutations in *BRAF* and *USP48* should be involved in the pathogenesis of corticotroph adenomas.

*BRAF* V600E represents a mutational hotspot in melanomas[15], craniopharyngioma[28], and multiple types of cancers[15,28–32], and its oncogenic activity has been extensively characterized[15,28–32]. In our study, an elevated kinase activity of *BRAF* V600E compared to wild-type *BRAF* was observed in corticotroph adenomas just like in other types of tumors, leading to activation of MAPK pathway and transactivation of POMC, which is the precursor of ACTH. Deregulated MAPK signaling is frequently observed in corticotroph adenomas, promoting ACTH production[33,34]. Previous studies have demonstrated that the *USP8* mutation causes corticotroph adenomas mainly through activating the EGFR–MAPK signal cascades[7,8]. Orphan nuclear receptor Nur77 acts as a mediator of corticotrophin-releasing hormone (CRH)-induced *POMC* transcription and plays an important role in the hypothalamic–pituitary–adrenal axis. In addition, *POMC* transcription can also be induced by the c-Jun/c-Fos heterodimer, which acts by binding to the canonical AP1-binding site in the first exon of *POMC*[20,21] (Supplementary Figure 4). Activation of the MAPK pathway could stimulate phosphorylation and transcriptional activation of the transcriptional regulators Nur77[22], c-Jun, and c-Fos[23]. Collectively, our results demonstrated that Nur77 and the c-Jun/c-Fos heterodimer mainly mediate the stimulatory effect of *BRAF* V600E on *POMC* transcription. We next sought to evaluate the effect of the BRAF inhibitor vemurafenib on ACTH secretion in primary human corticotroph adenomas. The results supported the potential efficacy of vemurafenib in the

**Table 1 Clinical and phenotypic features of patients with Cushing's disease**

| Characteristics | Wild-type ($n = 55$) | BRAF V600E ($n = 15$) | P value | USP48 mutation ($n = 21$) | P value |
|---|---|---|---|---|---|
| Gender | | | | | |
| Male | 16 | 4 | 0.856[a] | 3 | 0.187[a] |
| Female | 39 | 11 | | 18 | |
| Age at diagnosis | | | | | |
| Median | 36 | 36 | 0.769[b] | 41 | 0.5[b] |
| Interquartile range | 27–46 | 26–46 | | 26.5–49.5 | |
| Clinical course | | | | | |
| Median | 36 | 24 | 0.438[b] | 12 | 0.06[b] |
| Interquartile range | 12–84 | 8–60 | | 3.5–48 | |
| Diameter (mm) | | | | | |
| Median | 10 | 7.5 | 0.502[b] | 8.5 | 0.437[b] |
| Interquartile range | 5–22 | 3.5–22 | | 6.4–10 | |
| Midnight plasma corticotropin (ACTH) (pg/ml) | | | | | |
| Median | 60.4 | 140.5 | 0.023[b] | 48.45 | 0.307[b] |
| Interquartile range | 49.05–107.47 | 82.15–220.04 | | 31.57–92.63 | |
| Midnight serum cortisol (µg/dl) | | | | | |
| Median | 17.5 | 23.82 | 0.007[b] | 17.7 | 0.673[b] |
| Interquartile range | 14.34–28.05 | 21.82–39.10 | | 13.74–33.88 | |
| Urinary free cortisol (µg/24 h) | | | | | |
| Median | 510.97 | 488.94 | 0.936[b] | 415.66 | 0.505[b] |
| Interquartile range | 273.69–1133.15 | 405.37–1116.78 | | 215.28–894.86 | |
| Invasiveness | | | | | |
| Yes | 13 | 2 | 0.546[a] | 4 | 0.816[a] |
| No | 39 | 10 | | 14 | |
| Immediate postoperative biochemical remission (%) | | | | | |
| Yes | 24 | 6 | 0.951[a] | 13 | 0.328[a] |
| No | 25 | 6 | | 8 | |
| Postoperative recurrence (%) | | | | | |
| Yes | 2 | 1 | 0.937[a] | 0 | 0.224[a] |
| No | 9 | 4 | | 8 | |

[a]Values were determined as compared to wild-type (Pearson's $\chi^2$ test)
[b]Values were determined as compared to wild-type (Mann–Whitney $U$-test)

treatment of corticotroph adenomas with the *BRAF* V600E mutation. Of note, although both corticotroph adenomas and craniopharyngioma are pituitary tumors, and share some common clinical features, it is generally believed that they have different cellular origins. Considering the fact that *BRAF* V600E has been detected in many types of brain tumors, the observation that corticotroph adenomas and craniopharyngioma[28] have *BRAF* V600E appears not to be the evidence to support any connection between these tumor types.

Missense mutations of *USP48*, including M415I/V substitutions, were found at a very low frequency in several types of human cancers in the available cancer data sets, and its biological significance in tumors has not been fully described[35]. A luciferase report assay showed that *USP48* M415I/V promoted the *POMC* promoter activity. Moreover, the higher mRNA abundance of *POMC* in AtT-20 cells stably transfected with *USP48* M415I/V, further supports the notion that the M415I/V substitution causes corticotroph adenomas by up-regulating *POMC* transcription. Different from *BRAF* V600E, *USP48* mutation-mediated transcriptional activation of *POMC* appears not be linked to the activation of Erk1/2 or its downstream signaling cascade. In contrast, we provide the evidence to demonstrate that the NF-κB pathway is involved in *USP48*-mediated transcriptional activation of *POMC*. Several cytokines and hormones, including CRH, were reported to induce transcriptional activation of the pituitary *POMC* gene by directly suppressing NF-κB activity in AtT-20 cells[26]. AtT-20 cells displayed constitutive NF-κB activation, and inhibition of NF-κB activity increased transcription of the *POMC*

gene. Moreover, *USP48* has been implicated in the activation of NF-κB[24,25]. Accordingly, USP48-mediated inhibition of NF-κB activity could promote the transcription of *POMC* gene. However, these *USP48* mutants also promoted the activity of NF-κB binding site-mutated *POMC* promoters (Supplementary Figure 11), suggesting the existence of alternative USP48 substrates involved in the transcriptional control of *POMC*. It is very likely that *USP8* mutation activates the transcription of *POMC* via multiple pathways including the NF-κB pathway. Further studies are needed to reveal the underlying molecular mechanisms.

In summary, our study identified frequent *BRAF* and *USP48* mutations in corticotroph adenomas carrying wild-type *USP8* and indicated that these mutations cause Cushing's disease mainly by activating *POMC* gene transcription and increasing plasma ACTH levels (Fig. 3f). ACTH overproduction is a hallmark of Cushing's disease and appears to be frequently induced by mutations in genes that tightly regulate *POMC* gene transcription in the pathogenesis of this disease. This concept is further supported by the observation that except for *USP8*, *BRAF*, and *USP48*, there were detectable somatic mutations in genes that potentially affect MAPK signaling and *POMC* transcription. These mutated genes include serum response factor (*SRF*) and *PPP2R5D*, which were detected in individual tumor samples (Supplementary Data 1). Similar to the *USP8* mutation, *BRAF* V600E and *USP48* M415I/V were not detected in other types of pituitary adenomas, highlighting their specificity for corticotroph adenomas. Unlike malignant tumors, benign pituitary adenomas,

including corticotroph adenomas, have a relatively low number of somatic mutations per tumor, suggesting that each mutation found in a pituitary adenoma has a higher chance to be a driver mutation and might have a profound effect on its pathogenesis. The mutational status of *BRAF*, *USP8*, and *USP48* in corticotroph adenomas may be used in the future to characterize the molecular subtypes and guide targeted molecular therapy. Our findings suggest an immediately operable drug target for corticotroph adenomas carrying *BRAF* V600E.

## Methods

**Study patients**. Patients were recruited for this study at the Department of Neurosurgery in Huashan Hospital affiliated to Shanghai Medical College, Fudan University from 2003 to 2016. A total of 229 patients (91 with *USP8*-WT ACTH-secreting adenomas, 78 with *USP8*-mutated ACTH-secreting adenomas, 20 with growth hormone-secreting adenomas, 20 with prolactin-secreting adenomas, and 20 with non-functioning adenomas) were included in this study. This study was approved by the ethics committee of Huashan Hospital and informed consent was obtained from each patient. Cushing's disease was diagnosed according to the clinical manifestations, endocrine laboratory tests and imaging[8], and was histologically confirmed after surgical resection.

**Whole-exome sequencing**. A DNeasy Blood and Tissue DNA isolation kit (Qiagen) was used to extract genomic DNA from freshly frozen tumors and peripheral blood samples. An ultrasonicator Covaris E-220 (Covaris) was used to randomly fragment genomic DNA to 200–300 bps in size. We used Agencourt AMPure XP beads (Beckman) to purify the DNA and then ligated adaptors with sample-specific barcodes at both ends. The adapter-ligated products were purified using beads and amplified by ligation-mediated PCR. A TruSeq™ Exome Enrichment Kit and an Agilent SureSelect Human All Exon V5+UTR Kit (Agilent) was used to capture exome-enriched DNA fragments. Then, Agilent 2100 was used to estimate the magnitude of enrichment. Each library was loaded on the HiSeq 2500 platform for sequencing.

**Bioinformatic analysis**. Raw sequencing reads were filtered with trimmomatic (http://www.usadellab.org/cms/index.php?page=trimmomatic) and mapped to the reference hg19 human genome with Burrows-Wheeler Aligner (BWA) 0.7.7. PCR duplicates were removed with MarkDuplicates tool in Picard (http://picard.sourceforge.net), and then locally realigned using the Indel Realignment and recalibrated using the BaseRecalibrator tool from GATK. Somatic single-nucleotide variants (SNVs) were detected using MuTect v1.1.7 and VarScan 2, while somatic indels were identified using VarScan 2 and Genome Analysis Toolkit Somatic indel detector with default parameters. The set of mutations passed by somatic caller was dropped into post-processing filters to reduce false positive calls.

Filter criteria for high-confidence were as follows: (1) a minimum depth of 5× in both tumors and their normal pairs; (2) reads depth of variant alleles in tumors should be more than 4×, and with read quality of 20; (3) to avoid artifacts and likewise enable calling more variants on sites with lower error rates, somatic variants were filtered for allele frequencies greater than 10%. Filtered somatic mutations were manually reviewed for accuracy using Integrative Genomics Viewer (IGV, Broad Institute, Cambridge, MA, USA). All high-quality somatic mutations were annotated based on the information available in catalog of somatic mutations in cancer (COSMIC; http://cancer.sanger.ac.uk/cosmic), dbSNP138 (http://www.ncbi.nlm.nih.gov/snp), the 1000 Genomes Project (http://www.1000genomes.org/), and EXAC (http://exac.broadinstitute.org/) by Annovar. Germline variants with mutation frequencies >0.1% of the allelic fraction in these databases were removed. Mutational signatures in our study were identified by deconstructSigs[12].

**Candidate gene validation**. SNV and indels were validated by PCR reaction and Sanger sequencing. PCR primers were designed using the online software Primer3 (http://frodo.wi.mit.edu/primer3/) using GRCh37/hg19 as the reference sequence. We used a 96-well GeneAmp PCR System 9700 (Applied Biosystems) to perform the PCR reaction. A total of 10 ng of DNA from each sample was used per reaction with the 2× Taq PCR Master Mix (Vazyme Biotech). The sequencing reactions were performed with the Big Dye Terminator v.3.1 kit (Applied Biosystems). PCR products were sequenced using a 3730xl DNA Analyzer (Applied Biosystems).

**DNA constructs and site-directed mutagenesis**. Full-length wild-type cDNA of human *USP48* and *BRAF* were cloned into the lentivirus expression vector pCDH-EF1-MCS-T2A-Puro (System Biosciences). The desired mutations in *BRAF* and *USP48* cDNA were introduced using a QuikChange II Site-Directed Mutagenesis Kit (Agilent Technologies). The primers used in the construction of cDNA expression plasmids and mutagenesis are listed in Supplementary Table 3.

**Homology modeling**. A Blast search was used to search for homologs of USP48. The crystal structure of USP7 (PDB ID:1NBF[36]) ubiquitin aldehyde (ubal)-binding

form was used as the template to build the 3D structure of USP48 by "Build Homology Models" module of Discovery studio 4.0 software (Accelrys, Inc., San Diego, CA). The geometrical reasonability of the modeled USP48 structure was checked with a Procheck[37] analysis.

**Immunohistochemistry**. Consecutive sections (5 µm thick) of each formalin-fixed paraffin-embedded (FFPE) tissue block were subjected to immunohistochemical staining. Briefly, the slides were deparaffinized, re-hydrated, and then immersed in distilled water with 3% hydrogen peroxidase in methanol for 20 min to suppress endogenous peroxidase activity. Sections were incubated with primary antibodies of anti-USP48 (1:200, Abcam), anti-BRAF (1:200, Abcam), anti-p-Erk1/2 (1:200, Abcam), and anti-ACTH (1:200, Abcam) at 4 °C overnight. Subsequently, an ABC staining kit (Vector Laboratories) was applied to the sections, and signals were developed with diaminobenzidine (DAB) chromogen (Vector Laboratories). The analysis of our immunohistochemical stainings is based on the evaluation method according to ref.[38].

**Western blotting**. Cell and tumor tissues were lysed in RIPA lysis buffer (Sigma) supplemented with complete protease inhibitor cocktail (Roche). The resultant protein extracts were subjected to western blotting using a standard protocol. The following primary antibodies were used in this study: anti-actin (1:5000, Abcam); anti-USP48 (1:1000 Abcam); anti-GAPDH (1:1000, Abcam); anti-BRAF (1:1000, Abcam), anti-Erk1/2, anti-p-Erk1/2 anti-c-Fos (1:1000, Cell Signaling Technology), anti-c-Jun (1:1000, Cell Signaling Technology), anti-Nur77 (1:1000, Abcam), anti-p-c-Fos (1:1000, Cell Signaling Technology), anti-p-c-Jun (1:1000, Cell Signaling Technology), and anti-p-Nur77 (1:1000, Cell Signaling Technology). Uncropped images of the blots are provided in Supplementary Figure 12.

**Dual-luciferase reporter assay**. For luciferase reporter assays, AtT-20 cells seeded in a 24-well plate were transiently transfected with PGL3 basic vector (Promega), together with the transfer vector. Luciferase activity was detected using the Dual-luciferase reporter assay systems (Promega) according to the manufacturer's instructions. Chemiluminescence was measured using an EnVision Multilabel Reader (PerkinElmer, Waltham, MA, USA).

**Quantitative RT-PCR**. To determine the relative mRNA level of *POMC*, quantitative RT-PCR was performed as previously described[8]. RT-PCR was performed using SYBR® Premix Ex Taq™ (Tli RNaseH Plus) (TAKARA) to examine the mRNA abundance. Hypoxanthine-guanine phosphoribosyltransferase (*HPRT*) was used as an internal control to normalize the expression levels. The following primers were used: for POMC, forward 5′-GAGGCCACTGAACATCTTTGTC-3′, reverse 5′-GCAGAGGCAAACAAGATTGG-3′; for HPRT, forward 5′-ACCAGT CAACAGGGGACATAAA-3,′ reverse 5′-CTGACCAAGGAAAGCAAAGTCT-3′.

**Cell culture and transfection**. All cell lines used in this study were purchased from the American Type Culture Collection (ATCC, Manassas, VA, USA). The cell lines were routinely tested for mycoplasma contamination using the MycoAlert detection kit (Lonza). AtT-20 and HEK293 cell lines were maintained in DMEM (HyClone) supplemented with 10% fetal calf serum (FCS) at 37 °C in a humidified atmosphere with 5% $CO_2$. Methods of transfection and culture of primary tumor cells derived from human ACTH-secreting pituitary adenomas were the same as previously described[8]. Transfection was performed using SuperFect Transfection Reagent (QIAGEN) according to manufacturer's protocol. After 24 h of culture, primary human pituitary cells were incubated with the BRAF inhibitor vemurafenib (0, 0.3, 1, and 3 µM) for 48 h before analysis for ACTH secretion.

**Radioimmunoassay**. Plasma ACTH concentrations were measured by a sensitive and specific immunoradiometric assay. The sensitivity of the assay was 4.13 nmol/L for ACTH, and the inter- and intra-assay coefficients of variations were between 5–7% and 6–7.9%, respectively. The final results are expressed as picograms per milliliter. Cortisol concentrations were measured using a commercially available RIA kit (ICN Biomedicals, Inc., Costa Mesa, CA).

**Statistical analysis**. Statistical analysis was performed using SPSS version 17.0 (SPSS Inc.). Prism 6.0 (GraphPad) software was used to prepare graphs. Pearson's $\chi^2$ test was used for the categorical variables. The Mann–Whitney $U$-test was used for the continuous variables. $P$-values less than 0.05 were considered to be statistically significant.

**Data availability**. Sequence data have been deposited at the European Genome-phenome Archive (EGA), which is hosted by the EBI and the CRG, under accession number EGAS00001003029.

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

## Acknowledgements

We gratefully acknowledge Prof. Xingdang Liu for helpful discussions. We thank Mrs. Yun Zhang, Mrs. Qiuwei Song, and Prof. Yun Lu for sample collection. We also thank Mrs. Jingjing Zhu and Mr. Chao Li for their technical assistance with immunohisto-chemical staining. We thank Mrs. Xiaolan Qin and Ms. Jie Zheng, and the Genetron Health (Beijing) Co., Ltd. for technical support. This study was supported by the National Key Basic Research Program of China (973 Program) (2015CB559100), the National Natural Science Foundation of China (31325014, 81421061, 81501154), the Program of Shanghai Academic Research Leader (15XD1502200), the National Key R&D Program (2016YFC1201701), the National Key R&D Program-Special Project on Precision Medicine (2016YFC0903402), Shanghai Key Laboratory of Psychotic Dis-orders (13dz2260500), the Research Project of Shanghai Health and Family Planning Commission (201440552) to Y. Shi; 1000-Youth Elite Program and Shanghai Pujiang Scholar (grant 5PJ1407400) to C.H.; the China Pituitary Adenoma Specialist Council (CPASC), the National High Technology Research and Development Program of China (863 program; 2014AA020611), Chang Jiang Scholars Program, the National Program for Support of Top-Notch Young Professionals, the National Science Fund for Dis-tinguished Young Scholars (81725011), Shanghai Rising-Star Tracking Program (12QH1400400) to Y. Zhao.

## Author contributions

Y. Shi, C.H., and Y. Zhao were the overall principal investigators for the study who conceived the study and obtained financial support, and they were responsible for study design and supervised the entire study. J.C., X.J., and S.D. participated in the study design. Y. Zhao, X.S., Yongfei Wang, S.L., Z.Z., H. Ye, Y.L., H.C., F.T., Z. Yao, Y.M., and L.Z. supervised the diagnosis of patients and subject recruitment. Y. Zhao, Z.M., Y. Shen, Q.Z., Z. Ye, M.S., Y. Zhang, Z. Shi, C.C., and Ye Wang coordinated and provided samples. C.L., D.Z., and A.F. performed protein structure prediction. Y. Shi, Z. Song, Z.L., J.S., L.X., and C.F. performed biostatistics and bioinformatics analyses, and the results were interpreted by Y. Shi and H. Yan. X.J., C. Huang, J.C., S.D., H.P., C.P., D.G., M.C., and J.Z. performed the experiments. Y. Shi supervised the experiments and data analyses. The manuscript was drafted by J.C. and X.J. under the supervision of Y. Shi, C. Huang, and Y. Zhao who synthesized the manuscript. All authors critically reviewed the article and approved the final manuscript.

## Additional information

**Competing interests:** The authors declare no competing interests.

Jianhua Chen [1,2], Xuemin Jian[1,2], Siyu Deng[3], Zengyi Ma[4,5,6], Xuefei Shou[4,5,6], Yue Shen[4,5,6], Qilin Zhang[4,5,6], Zhijian Song[1,2], Zhiqiang Li [1,2], Hong Peng[3], Cheng Peng[3], Min Chen[7,8], Cheng Luo [9], Dan Zhao[8,9], Zhao Ye[4,5,6], Ming Shen[4,5,6], Yichao Zhang[4,5,6], Juan Zhou[1,2], Aamir Fahira[1,2], Yongfei Wang[4,5,6], Shiqi Li[4,5,6], Zhaoyun Zhang[5,10], Hongying Ye[5,10], Yiming Li[5,10], Jiawei Shen[1,2], Hong Chen[5,11], Feng Tang[5,11], Zhenwei Yao[5,12], Zhifeng Shi[4,5,6], Chunjui Chen[4,5,6], Lu Xie[13], Ye Wang[4,5,6], Chaowei Fu[14], Ying Mao[4,5,6,15], Liangfu Zhou[4,5,6], Daming Gao [7], Hai Yan[16], Yao Zhao[4,5,6,15], Chuanxin Huang[3] & Yongyong Shi [1,2,17,18,19]

[1]Shanghai Key Laboratory of Psychotic Disorders, Shanghai Mental Health Center, Shanghai Jiao Tong University School of Medicine; Bio-X Institutes, Key Laboratory for the Genetics of Developmental and Neuropsychiatric Disorders (Ministry of Education), and the Collaborative Innovation Center for Brain Science, Shanghai Jiao Tong University, Shanghai 200030, China. [2]Department of Otolaryngology Head and Neck Surgery & Center of Sleep Medicine, Shanghai Jiao Tong University Affiliated Sixth People's Hospital, Shanghai 200233, China. [3]Shanghai Institute of Immunology, Key Laboratory of Cell Differentiation and Apoptosis of Chinese Ministry of Education, Shanghai Jiao Tong University School of Medicine, Shanghai 200025, China. [4]Department of Neurosurgery, Huashan Hospital, Shanghai Medical College, Fudan University, Shanghai 200040, China. [5]Shanghai Pituitary Tumor Center, Shanghai 200040, China. [6]Institute of Neurosurgery, Fudan University, Shanghai 200040, China. [7]CAS Key Laboratory of Systems Biology, CAS Center for Excellence in Molecular Cell Science, Innovation Center for Cell Signaling Network, Shanghai Institute of Biochemistry and Cell Biology, Chinese Academy of Sciences, Shanghai 200031, China. [8]University of Chinese Academy of Sciences, Beijing 100049, China. [9]Drug Discovery and Design Center, State Key Laboratory of Drug Research, Shanghai Institute of Materia Medica, Chinese Academy of Sciences, Shanghai 201203, China. [10]Department of Endocrinology, Huashan Hospital, Shanghai Medical College, Fudan University, Shanghai 200040, China. [11]Department of Pathology, Huashan Hospital, Shanghai Medical College, Fudan University, Shanghai 200040, China. [12]Department of Radiology, Huashan Hospital, Shanghai Medical College, Fudan University, Shanghai 200040, China. [13]Shanghai Center for Bioinformation Technology (SCBIT), Shanghai Academy of Science and Technology, Shanghai 201203, China. [14]Department of Epidemiology, School of Public Health, Fudan University, Shanghai 200032, China. [15]State Key Laboratory of Medical Neurobiology, Institute of Neurosurgery, Shanghai Medical College, Fudan University, 200040 Shanghai, China. [16]Department of Pathology, Preston Robert Tisch Brain Tumor Center, Duke University Medical Center, Durham, NC 27710, USA. [17]Institute of Neuropsychiatric Science and Systems Biological Medicine, Shanghai Jiao Tong University, Shanghai 200030, China. [18]Department of Psychiatry, First Teaching Hospital of Xinjiang Medical University, Urumqi, Xinjiang 830054, China. [19]The Affiliated Hospital of Qingdao University & The Biomedical Sciences Institute of Qingdao University (Qingdao Branch of SJTU Bio-X Institutes), Qingdao University, Qingdao, Shandong 266003, China. These authors contributed equally: Jianhua Chen, Xuemin Jian, Siyu Deng, Zengyi Ma, Xuefei Shou, Yue Shen.

