## [Peer Review File · Nature Communications]

Reviewers' comments:

Reviewer #1 (Remarks to the Author):

The paper reports on the identification by whole exome sequencing (WES) of BRAF and USP48 somatic mutations in corticotroph adenomas without the recently identified USP8 mutations. An initial series of 22 USP8 wild type tumors have been sequenced by WES. Extending the series by targeted sequencing in a total series of 91 tumors the figures are: 16.5 % with BRAF V600E somatic mutation and 23 % USP48 mutants (M415I mutation in 11 cases and M415V mutation in 4 cases). Mutations of USP8, USP48 and BRAF are mostly mutually exclusive.

Studies supporting the role of these mutations in ACTH pituitary adenomas are:

- immunochemical analysis : corticotroph adenomas carrying the BRAF V600E mutation have higher MAPK 160 activity
- BRAF V600E expression induces phosphorylation of Nur77, c-jun and c-fos and stimulates the POMC promoter in the corticotroph AtT-171 20 cells
- Primary cultures of tumor cells with BRAF V600E were more sensitive to vemurafenib, showing new therapeutic perspectives.
- USP48 M415I/V mutants inhibits activity of NF- κ B reporter more than wild-type USP48, indicating gain-of-function mutation.
- USP48 M415I/V mutants stimulates POMC expression

The identification of BRAF and USP48 mutations in these tumors is original and could have, at least for BRAF, therapeutic consequences as demonstrated in others neoplasms. The study is well performed and on a large set of sample for a rare tumor.

Specific comments:

- BRAF and USP48 are clearly recurrent in this study. However were other genes altered in at least two samples ? A figure showing the frequency of each mutated gene derived from Sup table 1 would be interesting.
- It seems on Fig 1A that the signature of SNV in the BRAF mutated tumors differs from the one of the other tumors ? Are there any correlation between the signatures and the driver genes in this data set ?
- in Fig 2F only one mutated tumor was investigated. It is difficult to be convinced with n=1 ? Were other mutated tumors investigated ?
- Are there any clinical correlations (tumor size, Cushing syndrome, ACTH levels...) with the somatic mutations ? Are the BRAF tumors more aggressive ?

Minor misprints:

We screened the prevalence of USP48 and BRAF mutations in addition in an additional 69 corticotroph adenomas with wild-type USP8.

Taken together, these results indicate that BRAF 184 V600E contribute to the pathogenesis of ACTH-secreting Pas through increased 185 phosphorylation of POMC transcription regulators including Nur77, c-jun and c-fos.

Reviewer #2 (Remarks to the Author):

Manuscript by Chen et al. describes a genomic approach to identify genetic mutations responsible for Cushing's disease, a non-malignant pituitary tumor. This study is the extension of previous studies on genetics of the disease by the same group (and others), which had reported on frequent mutations of USP8 in these tumors (35-62% of the cases). The novelty, however, is the focus on USP8-wild type tumor aiming at identifying novel driver genes. Thus, authors have conducted whole-exome sequencing in 22 tumor-normal pairs with USP8-wild type tumors, and have identified recurrent mutations in two new genes; the well known BRAF V600E, and a hotspot (M415I/V) in USP48. Authors have validated these mutations using Sanger sequencing and have verified their recurrent status in a larger and independent sample set. Furthermore, the functionality of these new variants are examined in a pituitary cell line, by showing that these mutations leads to overexpression of POMC, which has previously been implicated in etiology of the disease.

Overall, the findings reported in this manuscript are novel, and advance our understanding of the genetic drivers of the disease, but perhaps the most interesting finding is the potential application of BRAF inhibitors for therapeutic intervention in subset of patients, i.e. patients with tumors affected by BRAFV600E mutation. Nevertheless, few points need to be addressed or further discussed in the manuscript, as I have described below;

- Authors have reported that "In a total 169 cases, USP8, USP48 and BRAF mutations were significantly mutually exclusive." What is the level of significance? Authors need to report a p-value, using an appropriate statistical test, for this observation.

- Authors report that "We observed a preference for C>T/G>A alterations analogous to the somatic single-nucleotide variation (SNV) spectrum (Fig. 1A)". There are known mutational signatures associated with this pattern. Have authors compared their results to known mutational signatures? Specifically, is the observed C>T mutations associated with the presence of cytosines in the context of CpG dinucleotides? (which have been reported to be target of deamination when methylated). Adding a paragraph to the Discussion about this observation, and possible reasons would be helpful.

-Discussion can be improved by comparing results to other studies and other related tumor types. For example BRAFV600E mutation has been observed in papillary craniopharyngiomas, another type of pituitary tumors (PMID: 28486603). Can authors discuss this and explore possible connection between these tumor types?

- Methods for bioinformatic analysis and identification of somatic mutations should be described.
- Source of the cell lines used in this study should be mentioned.

Reviewer #3 (Remarks to the Author):

In this manuscript, Chen et al. performed whole-exome sequencing plus sanger sequencing validation in 169 patients with Cushing's disease, and identified 2 novel recurrently mutated genes, USP48 (M415I or M415V, in 22 cases) and BRAF (V600E, in 19 cases), which are mutually exclusive with previously reported USP8 mutation and absent in other types of pituitary adenomas. Either USP48 or BRAF mutations could enhance the POMC transcription, contributing to the ACTH overproduction. Of great clinical significance, corticotroph tumor cells harboring BRAF V600E were much more sensitive to the BRAF inhibitor vemurafenib, providing insights into the therapeutics of this disease. This is a straightforward research that is well designed and well executed. The manuscript is also well written with clear presentation and description. The data is of high quality and strongly supports the overall conclusion of this study. Moreover, there are noted weaknesses in this study. The manuscript can be further improved if the following concerns are addressed.

1. Clinical relevance would be strengthened if the authors include a comparison of clinical characteristics of patients such as age, gender, and tumor size, plasma ACTH with or without USP48/BRAF/USP8 mutations. Previous analysis demonstrated that USP8 mutations usually appear in young adult females, which indicated the effect of sex hormones in the pathogenesis of USP8 mutated tumors. It would be significant if there were a similar tendency for USP48/BRAF mutations.
2. In line 141, the authors mentioned, "Among the 21 USP48-mutated cases, 11 cases had a M415I mutation and 4 cases harbored a M415V mutation". What are the mutations in other 6 cases? Could the authors provide a list including all the mutation forms they observed?
3. Data of Figure 2C was indicated to correspond to Figure 2A, AtT-20 cells. However, a similar IB analysis on these signaling proteins should be included in Figure 2 with at least four tissue samples, normal pituitary, and tumors with USP8 mutation, USP48 mutation, and BRAF/V600E.
4. In Figure 2F, the authors showed different vemurafenib sensitivities in 2 BRAF-WT and only 1 BRAF V600E corticotroph adenomas. Since it is a key data to show the therapeutic significance, could the authors add more data from different BRAF V600E tumors? Moreover, the authors should show the survival of tumor cells under vemurafenib treatment to elucidate whether the decreased ACTH productivity is on account of transcriptional regulation or decreased tumor cell number.
5. In Figure 3, the authors used molecular modeling to predict that USP48M415V associates/binds to NF- κ B as depicted in Figure 3F, affecting NF- κ B-responsive element. Although promoter assays support this hypothesis (Figure 3C), reciprocal IP-IB experiments must be performed to confirm this speculation. Additionally, the effects of exogenous expression of WT, M451I or M415V USP48 with BRAF V600E as a control on ACTH secretion in AtT-20 cells should also be examined.

6. Since NF- κ B is not involved in the USP48-mediated transcriptional control of POMC, did authors try to explore other mechanism? For instance, the authors can compare the level of p-ERK, p-Nur77, p-c-FOS and p-c-JUN in USP48-WT or M415I/V-overexpressing cells, and detect the effect of USP48 mutants on the activities of NurPENBRE/AP1-mutated POMC promoters.

7. The immunohistochemical staining in supplementary Figure 5 seems the expression level of USP48 protein is a little higher in USP48 wild-type tumors, which is opposite to the conclusion of line 188-191. Besides, the author should have a statistical data to support their conclusion.

8. For readers who are not familiar with pituitary tumors, it would be beneficial the authors provide more information about this tumor in the introduction such as frequency of tumor occurrence, current therapy option and prognosis of this tumor.

9. English editing is needed for grammar and spelling errors. For examples, in line 219, delete "in additional"; in line 223, missing "." after "PAs"; in line 252, missing "." after "ArT-20 cells".

Point-by-point response for NCOMMS-17-18720A

We would like to thank the editors and the three reviewers for their positive and encouraging comments as well as their constructive criticisms. We have performed experiments and introduced improvements in the manuscript to address each of these questions. We realize and appreciate that these suggestions have significantly strengthened the manuscript and hence we are truly grateful for the excellent input of all three reviewers. The changes we have made are highlighted by **blue fonts** in the manuscript.

Reviewers' comments:

Reviewer #1 (Remarks to the Author):

The paper reports on the identification by whole exome sequencing (WES) of BRAF and USP48 somatic mutations in corticotroph adenomas without the recently identified USP8 mutations. An initial series of 22 USP8 wild type tumors have been sequenced by WES. Extending the series by targeted sequencing in a total series of 91 tumors the figures are: 16.5 % with BRAF V600E somatic mutation and 23 % USP48 mutants (M415I mutation in 11 cases and M415V mutation in 4 cases). Mutations of USP8, USP48 and BRAF are mostly mutually exclusive.

Studies supporting the role of these mutations in ACTH pituitary adenomas are:

- immunochemical analysis : corticotroph adenomas carrying the BRAF V600E mutation have higher MAPK 160 activity-BRAF V600E expression induces phosphorylation of Nur77, c-jun and c-fos and stimulates the POMC promoter in the corticotroph AtT-171 20 cells
- Primary cultures of tumor cells with BRAF V600E were more sensitive to vemurafenib, showing new therapeutic perspectives.
- USP48 M415I/V mutants inhibits activity of NF-κB reporter more than wild-type USP48, indicating gain-of-function mutation.
- USP48 M415I/V mutants stimulates POMC expression

The identification of BRAF and USP48 mutations in these tumors is original and could have, at least for BRAF, therapeutic consequences as demonstrated in others neoplasms. The study is well performed and on a large set of sample for a rare tumor.

Specific comments:

1.- BRAF and USP48 are clearly recurrent in this study. However were other genes altered in at least two samples? A figure showing the frequency of each mutated gene derived from Sup table 1 would be interesting.

Answer: We thank the reviewer for this comment. In addition to *BRAF* and *USP48*, *NR3C1* and *HCFC1* were found to be mutated in at least two cases of the 22 corticotroph adenomas in this study. As the reviewer suggested, we added the table with frequencies of each recurrently mutated gene in 22 corticotroph adenomas (Supplementary Table 2**). (Page 5, Line 141-145)**

2.It seems on Fig 1A that the signature of SNV in the BRAF mutated tumors differs from the one of the other tumors? Are there any correlation between the signatures and the driver genes in this data set ?

Answer: We thank the reviewer for this comment. We also noted that *BRAF* mutated tumors (#7 and #8) had the A>T alteration, but this mutation existed in only five out of the 20 *BRAF* wild-type tumors. In fact, both sample #7 and #8 only have one site carrying the A>T mutation.

Corticotroph adenoma is a benign tumor and has a low number of somatic mutations per case (ranging from 2 to 34). Based on our current sequencing data, we cannot conclude that *BRAF* mutated tumors have unique SNV signature. Moreover, we cannot find any correlation between the signatures and the driver genes.

3.-in Fig 2F only one mutated tumor was investigated. It is difficult to be convinced with n=1 ? Were other mutated tumors investigated ?

Answer: We completely agree with the reviewer. In fact, the third reviewer has the similar comment. We tested the effect of vemurafenib on ACTH production in another primary cultured human corticotroph adenoma with *BRAF* mutation. We found that this tumor was sensitive to vemurafenib as well. This result further consolidates our previous conclusion. We have also modified Fig. 2f accordingly. Due to the difficulty of obtaining fresh *BRAF*-mutated tumors for culture, only one more *BRAF*-mutated tumor was obtained for our study when we revised this manuscript. We are continuing to work on this as we also think this is important for the development of potential target therapy for patients carrying *BRAF*-mutated tumors.

4.- Are there any clinical correlations (tumor size, Cushing syndrome, ACTH levels...) with the somatic mutations ? Are the BRAF tumors more aggressive ?

Answer: We thank the reviewer for raising the interesting questions. We compared the clinical features of patients with different mutation status (**Table 1**). The patients with mutated *BRAF* had significantly higher levels of midnight plasma corticotrophin ($P = 0.023$) and midnight serum cortisol ($P = 0.007$) levels compared to those with wild-type *BRAF* (**Table 1**). Interestingly, *BRAF* mutated tumors displayed similar sizes as wild-type tumors, indicating that *BRAF* mutation promotes ACTH production. *BRAF*-mutated tumors are not more aggressive when compared with wild-type tumors (**Table 1**). Patients carrying *USP48* mutation do not display any significant difference in all tested clinical features than patients with wild-type *USP48*. Accordingly, we included these results in the revised manuscript. (Page 7, Line 238-247)

Minor misprints:

We screened the prevalence of *USP48* and *BRAF* mutations in additional in an additional 69 corticotroph adenomas with wild-type *USP8*.

Taken together, these results indicate that *BRAF* 184 V600E contribute to the pathogenesis of ACTH-secreting Pas through increased 185 phosphorylation of POMC transcription regulators including Nur77, c-jun and c-fos.

Answer: Typo has been corrected in the revised manuscript.

Reviewer #2 (Remarks to the Author):

Manuscript by Chen et al. describes a genomic approach to identify genetic mutations responsible for Cushing's disease, a non-malignant pituitary tumor. This study is the extension of previous studies on genetics of the disease by the same group (and others), which had reported on frequent mutations of *USP8* in these tumors (35-62% of the cases). The novelty, however, is the focus on *USP8*-wild type tumor aiming at identifying novel driver genes. Thus, authors have conducted whole-exome sequencing in 22 tumor-normal pairs with *USP8*-wild

type tumors, and have identified recurrent mutations in two new genes; the well known BRAF V600E, and a hotspot (M415I/V) in USP48. Authors have validated these mutations using Sanger sequencing and have verified their recurrent status in a larger and independent sample set. Furthermore, the functionality of these new variants are examined in a pituitary cell line, by showing that these mutations leads to overexpression of POMC, which has previously been implicated in etiology of the disease.

Overall, the findings reported in this manuscript are novel, and advance our understanding of the genetic drivers of the disease, but perhaps the most interesting finding is the potential application of BRAF inhibitors for therapeutic intervention in subset of patients, i.e. patients with tumors affected by BRAF V600E mutation. Nevertheless, few points need to be addressed or further discussed in the manuscript, as I have described below;

1.- Authors have reported that "In a total 169 cases, USP8, USP48 and BRAF mutations were significantly mutually exclusive." What is the level of significance? Authors need to report a p-value, using an appropriate statistical test, for this observation.

Answer: We are sorry for this misleading statement and have clarified this issue in the revised manuscript. In detail, *USP8* mutation is significantly exclusive from mutations in either *USP48* or *BRAF* in a total of 169 cases ($P = 1.208 \times 10^{-5}$ and $P = 0.027$ receptively, two-sided Fisher's exact test). However, *USP48* and *BRAF* mutations are not mutually exclusive ($P > 0.05$, two-sided Fisher's exact test). We have modified the text accordingly. (Page 5, Line 157-160)

2.- Authors report that "We observed a preference for C>T/G>A alterations analogous to the somatic single-nucleotide variation (SNV) spectrum (Fig. 1A)". There are known mutational signatures associated with this pattern. Have authors compared their results to known mutational signatures? Specifically, is the observed C>T mutations associated with the presence of cytosines in the context of CpG dinucleotides? (which have been reported to be target of deamination when methylated). Adding a paragraph to the Discussion about this observation, and possible reasons would be helpful.

Answer: We used `deconstructSigs`¹ to identify mutational signatures of our samples. Mutation Signature 1A (**Figure R1**) is associated with the SNV spectrum found in these tumors.

Figure R1. Mutational signatures of our samples

Signature 1A is characterized by the prominence of C>T substitutions at the NpCpG trinucleotides². Indeed, some C>T mutations in these tumors occurred in the CpG sites. It is

well-known that CpG dinucleotide is the main mutational hot-spot in most human cancers. The possible reason is that the C of CpG is a preferential site for DNA methylation, and methyl-C is prone to spontaneous deamination to T³. This observation further supports this notion. In the revised manuscript we have added this result (Page 4, Line 132-136) and discussed this accordingly. (Page 7, Line 255-259)

3.-Discussion can be improved by comparing results to other studies and other related tumor types. For example BRAF V600E mutation has been observed in papillary craniopharyngiomas, another type of pituitary tumors (PMID: 28486603). Can authors discuss this and explore possible connection between these tumor types?

Answer: We thank the reviewer for this important comment. Indeed, pituitary adenoma and craniopharyngiomas are the most frequently types of tumors that can arise in the sellar and parasellar areas. Moreover, they have several similar clinical features. However, they differ from each other in many aspects including cellular origins. *BRAF* V600E mutation occurs in 95% of papillary craniopharyngiomas, rare types of craniopharyngiomas⁴, and in about 11% of corticotroph adenomas (19/169) based on our study. *BRAF* V600E mutation also occurs in many types of brain tumors including glioblastoma. It is common that the same gene mutation occurs in different types of human cancers. Therefore, our finding seems not to provide sufficient evidence to demonstrate any connection between these tumor types. We have stated this in the discussion section. (Page 8, Line 291-297)

4.Methods for bioinformatic analysis and identification of somatic mutations should be described.

Answer: Following the reviewer's advice, we have added the descriptions about the methods for bioinformatic analysis and identification of somatic mutations. (Page 15, Line 539-558)

5. Source of the cell lines used in this study should be mentioned.

Answer: All cell lines used in this study were purchased from the American Type Culture Collection (ATCC, Manassas, VA, USA). I have stated this in the revised manuscript. (Page 17, Line 620-621)

Reviewer #3 (Remarks to the Author):

In this manuscript, Chen et al. performed whole-exome sequencing plus sanger sequencing validation in 169 patients with Cushing's disease, and identified 2 novel recurrently mutated genes, USP48 (M415I or M415V, in 22 cases) and BRAF (V600E, in 19 cases), which are mutually exclusive with previously reported USP8 mutation and absent in other types of pituitary adenomas. Either USP48 or BRAF mutations could enhance the POMC transcription, contributing to the ACTH overproduction. Of great clinical significance, corticotroph tumor cells harboring BRAF V600E were much more sensitive to the BRAF inhibitor vemurafenib, providing insights into the therapeutics of this disease. This is a straightforward research that is well designed and well executed. The manuscript is also well written with clear presentation and description. The data is of high quality and strongly supports the overall conclusion of this study. Moreover, there are noted weaknesses in this study. The manuscript can be further improved if the following concerns are addressed.

1. Clinical relevance would be strengthened if the authors include a comparison of clinical characteristics of patients such as age, gender, and tumor size, plasma ACTH with or without USP48/BRAF/USP8 mutations. Previous analysis demonstrated that USP8 mutations usually appear in young adult females, which indicated the effect of sex hormones in the pathogenesis of USP8 mutated tumors. It would be significant if there were a similar tendency for USP48/BRAF mutations.

Answer: We thank the reviewer for these constructive suggestions. Following the reviewer's suggestion, we compared the clinical characteristics of patients with different mutation status (**Table 1**). As compared to patients with wild-type *BRAF/USP48*, patients with mutated *BRAF* had significantly higher levels of midnight plasma ACTH ($P = 0.023$) and midnight serum cortisol ($P = 0.007$) (**Table 1**), but similar tumor sizes. Therefore, *BRAF* mutation appears to promote ACTH production. Other clinical features examined are undistinguishable between patients with wild-type and mutated *BRAF*. Patients with *USP48* mutated tumors have quite similar clinical features as those with wild-type *BRAF/USP48*. There is no significant difference in gender and age among the three groups of patients. We have included this result accordingly in the revised manuscript. (Page 7, Line 238-247)

2. In line 141, the authors mentioned, "Among the 21 USP48-mutated cases, 11 cases had a M415I mutation and 4 cases harbored a M415V mutation". What are the mutations in other 6 cases? Could the authors provide a list including all the mutation forms they observed?

Answer: We have provided a table to describe all the *USP48* mutations we detected in the 169 cases (**Supplementary Table 3**).

3. Data of Figure 2C was indicated to correspond to Figure 2A, AtT-20 cells. However, a similar IB analysis on these signaling proteins should be included in Figure 2 with at least four tissue samples, normal pituitary, and tumors with USP8 mutation, USP48 mutation, and BRAF/V600E.

Answer: We think this is a great point. However, we cannot get normal pituitary for Western blot analysis. Instead, we performed immunohistochemical analyses for the expression levels of these signaling proteins. Consistent with the results obtained from AtT-20 (**Fig. 2c**), *BRAF*-mutated tumors are strongly positive for p-Nur77, p-c-Jun and p-c-Fos (**Supplementary Fig. 5**). *USP8*-mutated tumors are positive for p-Erk1/2, p-c-Jun and p-c-Fos. However, normal pituitaries and tumors with *USP48* mutation are negative or weakly positive for these molecules.

As suggested by the reviewer in Question 6, overexpression of *USP48* mutants cannot up-regulate the phosphorylation of these proteins in AtT-20 cells (**Supplementary Fig. 8**). Collectively, these results further support the notion that *BRAF* mutation, but not *USP48* mutation, promotes ACTH production via phosphorylation of Nur77, c-Jun and c-Fos in corticotroph adenomas. (Page 6, Line 215-224)

4. In Figure 2F, the authors showed different vemurafenib sensitivities in 2 BRAF-WT and only 1 BRAF V600E corticotroph adenomas. Since it is a key data to show the therapeutic significance, could the authors add more data from different BRAF V600E tumors? Moreover, the authors should show the survival of tumor cells under vemurafenib treatment to elucidate

whether the decreased ACTH productivity is on account of transcriptional regulation or decreased tumor cell number.

Answer: We completely agree with the reviewer. We tested the effect of vemurafenib on ACTH production in another human corticotroph adenoma primary cell culture carrying *BRAF* mutation. Vemurafenib treatment led to much more reduction of ACTH secretion in *BRAF* mutated tumor than wild-type. However, treatment with vemurafenib did not cause more cell death in *BRAF* mutated primary cultures (**Figure R2**), indicating that decreased ACTH levels were not caused by more cell death in *BRAF* mutated tumor. This result further consolidates our conclusion. We have added this point in the revised manuscript (**Page 6, Line 199-200**) and also modified Fig. 2f accordingly. Due to the difficulty in obtaining fresh *BRAF*-mutated tumor for culture, only one more *BRAF*-mutated case was obtained for our study when we revised this manuscript. We are continuing to work on this as we also think this is important for the development of potential target therapy for patients carrying *BRAF*-mutated tumors.

Figure R2. Human corticotroph adenoma primary cell cultures carrying wild-type or mutated *BRAF* were treated with vemurafenib at different doses for 48 h. The viable cells were determined by exclusion of PI.

5. In Figure 3, the authors used molecular modeling to predict that USP48M415V associates/binds to NF- κ B as depicted in Figure 3F, affecting NF- κ B-responsive element. Although promoter assays support this hypothesis (Figure 3C), reciprocal IP-IB experiments must be performed to confirm this speculation. Additionally, the effects of exogenous expression of WT, M451I or M415V USP48 with BRAFV600E as a control on ACTH secretion in AtT-20 cells should also be examined.

Answer: We have performed reciprocal IP-IB experiments to confirm that *USP48* mutants harbor the ability to interact with RelA, a subunit of NF- κ B (**Supplementary Fig. 9**). With respect to examining ACTH secretion in AtT-20 cells transfected with various constructs, unfortunately we did not find any difference. We measured human ACTH concentration in plasma from patients or culture medium using a sensitive and specific immunoradiometric assay. However, we have not established a similar method to measure mouse ACTH yet. Instead, we used mouse ACTH ELISA kit to determine ACTH levels. This assay displayed very poor dilution linearity and cannot differentiate the two samples. Therefore, we cannot draw any conclusion so far. We apologize for the delay in establishment of another sensitive method but fully appreciate the importance of this suggestion.

6. Since NF- κ B is not involved in the USP48-mediated transcriptional control of POMC, did authors try to explore other mechanism? For instance, the authors can compare the level of p-ERK, p-Nur77, p-c-FOS and p-c-JUN in USP48-WT or M415I/V-overexpressing cells, and

detect the effect of USP48 mutants on the activities of NurPENBRE/AP1-mutated POMC promoters.

Answer: We thank the reviewers for these valuable advices. Overexpression of *USP48* mutants failed to up-regulate the phosphorylation of Erk1/2, Nur77, c-Fos and c-Jun in AtT-20 cells (**Supplementary Fig. 8**). (Page 6, Line 215-219) *USP48*-mutated tumor cells were weakly or negatively stained for p-Erk1/2, p-Nur77, p-c-Fos and p-c-Jun (**Supplementary Fig. 5**). (Page 6, Line 219-221) Moreover, luciferase reporter assay showed that *USP48* mutants could still increase the activities of NurRE-, NBRE-, or AP1-mutated *POMC* promoters (**Figure R3**). Therefore, these signaling pathways appear not be involved in the *USP48*-mediated transcriptional control of *POMC*. We hypothesize that *USP48*-mediated transcriptional control of *POMC* involves multiple molecules including NF- κ B. We have stated this in the revised manuscript. (Page 8, Line 300-308)

7. The immunohistochemical staining in supplementary Figure 5 seems the expression level of USP48 protein is a little higher in USP48 wild-type tumors, which is opposite to the conclusion of line 188-191. Besides, the author should have a statistical data to support their conclusion.

Answer: We performed the immunohistochemical staining of USP48 in three wild-type and three *USP48*-mutated tumor specimens, and conducted quantitative analysis of USP48 protein levels in these samples. Our results demonstrate that *USP48*-mutated tumors have similar expression levels of USP48 protein as wild-type tumors (**Supplementary Fig. 6**).

8. For readers who are not familiar with pituitary tumors, it would be beneficial the authors provide more information about this tumor in the introduction such as frequency of tumor occurrence, current therapy option and prognosis of this tumor.

Answer: We totally agree. We have re-written the introduction section to include this information in the revised manuscript.

9. English editing is needed for grammar and spelling errors. For examples, in line 219, delete "in additional"; in line 223, missing "." after "PAs"; in line 252, missing "." after "ArT-20 cells".

Answer: We apologize for these errors. We have corrected the mistakes and asked a native speaker to polish our manuscript.

REFERENCE

1. Rosenthal R, McGranahan N, Herrero J, Taylor BS, Swanton C. DeconstructSigs: delineating mutational processes in single tumors distinguishes DNA repair deficiencies and patterns of carcinoma evolution. *Genome Bio***17**, 31 (2016).
2. Alexandrov LB, *et al.* Signatures of mutational processes in human cancer. *Nature***500**, 415-421 (2013).
3. Walser JC, Ponger L, Furano AV. CpG dinucleotides and the mutation rate of non-CpG DNA. *Genome Res***18**, 1403-1414 (2008).
4. Brastianos PK, *et al.* Exome sequencing identifies BRAF mutations in papillary craniopharyngiomas. *Nat Genet***46**, 161-165 (2014).

REVIEWERS' COMMENTS:

Reviewer #1 (Remarks to the Author):

The answers to my initial comments are acceptable

Reviewer #3 (Remarks to the Author):

The revised manuscript addressed all the concerns from this reviewer and the other two reviewers with additional data and necessary revision in the text. I think that this manuscript is ready for its publication.

NO further comments and concerns.

Point-by-point response for NCOMMS-17-18720A

We would like to thank the editors and the reviewers for their positive and encouraging comments. We revised the manuscript using the 'track changes' feature in MS-Word.

Reviewers' comments:

Reviewer #1 (Remarks to the Author):

The answers to my initial comments are acceptable

Response: Thanks for your comment and all previous valuable advices.

Reviewer #3 (Remarks to the Author):

The revised manuscript addressed all the concerns from this reviewer and the other two reviewers with additional data and necessary revision in the text. I think that this manuscript is ready for its publication.

NO further comments and concerns.

Response: Thank you for these positive and constructive comments which have greatly improved the quality of our manuscript.